# Riding a Mechanical Scooter from the Inconvenient Side Promotes Muscular Balance Development in Children

**DOI:** 10.3390/children10061064

**Published:** 2023-06-15

**Authors:** Mantas Mickevicius, Danguole Satkunskiene, Saule Sipaviciene, Sigitas Kamandulis

**Affiliations:** 1Institute of Sports Science and Innovations, Lithuanian Sports University, 44221 Kaunas, Lithuania; mantas.mickevicius@lsu.lt (M.M.); danguole.satkunskiene@lsu.lt (D.S.); 2Department of Health Promotion and Rehabilitation, Lithuanian Sports University, 44221 Kaunas, Lithuania; saule.sipaviciene@lsu.lt

**Keywords:** physical activity, muscle balance, symmetry index, opposite-side exercise, lower limbs

## Abstract

Mechanical scooter riding is a popular physical activity among children, but little is known about the differences in muscle loading between the dominant and non-dominant sides during this activity. The objective of this study was to identify the muscle activation patterns in children’s dominant and non-dominant legs as they rode scooters on the convenient and inconvenient sides. The study included nine healthy children aged 6–8. The participants rode 20 m on a mechanical scooter at a self-selected pace using both the convenient and inconvenient sides. Electromyography was used to measure the muscle activity in the dominant and non-dominant legs during the pushing and gliding phases. A 20 m sprint run was used as a control exercise to estimate the typical differences in muscle activation between the dominant and non-dominant legs. In the pushing phase, the symmetry index for five of the eight analyzed muscles exceeded 50% (*p* < 0.05); four of these muscles were more active in the pushing leg, and one was more active in the standing leg. In the gliding phase, four muscles were more active in the standing leg, and one was more active in the pushing leg (*p* < 0.05). Upon observing children who changed sides while riding a scooter, it was found that the pattern of muscle activation displayed a reverse trend that resembled the initial pattern. Our study indicated notable differences in muscle activity patterns between the dominant and non-dominant sides of individual leg muscles during children’s scooter riding. These patterns were reversed when children switched sides on the scooter. These findings suggest that using both legs and switching sides while riding a scooter may be a viable strategy for promoting balanced muscular development.

## 1. Introduction

A daily 60-min physical activity regimen is practiced by only 24% of the population aged 6 to 17 [1]. This is discouraging because participation in sports and physical activity has a favorable impact on health and quality of life, so initiatives that increase participation are supported. The primary driving factors behind children’s involvement in physical activity are often attributed to the concepts of “fun” and “enjoyment” [2]. Depending on the region, children and adolescents exhibit distinct preferences for participation, with team sports and swimming emerging as the predominant choices on a global scale [3]. Sports equipment such as bicycles, skateboards, rollerblades, and scooters considerably contribute to sustaining physical activity among youth because of the enjoyment, and this type of physical activity allows children and parents to spend their free time outdoors [4,5]. Currently, scooters are one of the most popular types of outdoor equipment and include electric scooters, non-motorized two-wheeled and three-wheeled scooters, and trick scooters for performing acrobatics and jumps in obstacle parks. Non-motorized two-wheelers are used widely from a young age because they are easier to ride than other types of scooters.

Inappropriate forms of physical activity can cause muscle imbalance and changes in the mechanical properties of tendons [6,7]. Muscle imbalance is a major cause of many disturbances in the musculoskeletal system [8,9]. For example, imbalance between the vastus medialis oblique and vastus lateralis (VL) muscles is a major factor in the development of patellofemoral pain syndrome [10]. Riding a two-wheeled scooter involves repetitive activity in which only one foot pushes off while the other remains fixed on the scooter. When regularly performed, this motion may cause an imbalance in the strength and mass of the leg muscles between the two sides of the body.

Asymmetry between the limbs used for an exercise reflects differences in muscle activation [11,12]. The demand of a greater load on one side of the body results in an increase in the electromyography (EMG) amplitude for the muscles on that side. Studies using EMG reported muscle imbalance during various exercises or movements under different conditions [13,14,15,16]. However, few studies focused on the activity of muscles in children while riding a scooter, specifically on the differences between the two sides of the body, between riding phases, and according to riding intensity. It may be postulated that using both sides of the body when riding a non-motorized scooter may reduce the likelihood of muscular imbalance, but it is not known whether doing so is appropriate and safe.

The purpose of this study was to identify the muscle activation patterns in children’s dominant and non-dominant legs as they rode scooters on the convenient and inconvenient sides. It was expected that the differences in muscle recruitment between the legs would be evident but would be totally compensated for while driving the scooter from the opposite side. This assumption is based on a long-standing strategy that advocates the implementation of unilateral exercises on the weaker side to minimize the asymmetry between limbs [17,18]. A running sprint exercise was used as a control to eliminate any bias resulting from the typical differences in muscle activation between the dominant and non-dominant legs.

## 2. Materials and Methods

### 2.1. Participants

The study involved nine healthy children aged 6–8. The participants’ age, gender, and anthropometric characteristics are shown in Table 1. The children possessed a range of two to four years’ experience in scooter riding. The criteria for inclusion in the study were no injuries of the musculoskeletal system or no history of orthopedic surgery and the basic skill to ride a scooter. The personal scooter of each child was used in the study. Six of the nine participants preferred to ride with their right leg on the scooter. The leg on the scooter was defined as the dominant leg. The convenient side was considered when driving with the dominant leg on the scooter. All procedures were conducted at the Institute of Sport Science and Innovations of Lithuanian Sports University. Participants were informed of the testing requirements and provided written informed consent. The study was approved by the Ethics Committee of the Lithuanian Sports University (No. MNLKIN(M)-2021-374, approved 18 March 2021) and followed the principles of the Declaration of Helsinki.

### 2.2. Testing Procedures

Each child first performed a 5-min warm up comprising dynamic stretching exercises for the lower body. EMG and inertial measurement units (IMU) sensors were then placed. Each child then completed four 20 m rides on the scooter at a self-selected speed, after which the participants performed two 20 m running sprints. One video camera was synchronized with the EMG and IMU records, and it was used to record the sprint running and scooter riding.

#### 2.2.1. 20 m Sprint Test

The 20 m linear sprint test was used as previously described [19,20]. The participants sprinted from a standing start position, and the time was recorded using a Witty timer system (Mahopac, NY, USA). Two trials were performed with about 3 min of passive recovery between them. The fastest sprint time was used for the analyses. The test–retest reliability of this procedure in young participants was previously reported (intraclass correlation = 0.95) [19].

#### 2.2.2. 20 m Ride Test

Each participant rode the scooter for 20 m at a self-paced effort; this was performed four times in total with 3–5 min rest between them. Two trials were performed from the convenient side (using the dominant leg on the scooter), and then two trials were performed from the inconvenient side (using the non-dominant leg on the scooter). The better result of each of the two trials for each leg was used for further analysis. The participants performed two trials with each leg on the scooter as a warm up just before the experimental rides. The start and end of 20 m distance were clearly marked by a 5 cm wide tape. The photosensors of the Witty timer system were used to measure riding time. All trials were performed from a stationary standing position starting 70 cm before the first photosensing element.

#### 2.2.3. Muscle Activity Recording

A Noraxon Ultium EMG sensor system (Noraxon MR3 3.18.18; Scottsdale, AZ, USA) was used for recording the EMG, which was sampled at 2000 Hz, and recorded for five muscles in the left and right thigh and three muscles in the left and right calf. The EMG was synchronized with the recording of the hip, knee, and ankle joint angles. One video camera (Nixon 125; Konan, Minato-ku, Tokyo) was sampled at 60 Hz and was synchronized with the EMG (Noraxon Ultium) and IMU (Research PRO IMU, Noraxon, Scottsdale, AZ, USA) to videotape the scooter rides.

Each participant’s skin over the target muscles was cleaned with an alcohol-soaked pad to reduce skin impedance, and disposable dual Ag-AgCl surface electrodes (Noraxon) were placed over the belly of the following muscles: (1) rectus femoris (RF) at 50% of the distance from the anterior spina iliaca superior to the superior aspect of the patella; (2) biceps femoris (BF) at 50% of the distance between the ischial tuberosity and the lateral epicondyle of the tibia; (3) semitendinosus (SM) at 50% of the distance between the ischial tuberosity and the medial epicondyle of the tibia; (4) tibialis anterior (TA) at one-third of the distance from the tip of the fibula to the tip of the medial malleolus; (5) gastrocnemius lateralis (GL) at one-third of the distance between the head of the fibula and the heel; (6) gastrocnemius medialis (GM) at the most prominent bulge of the muscle; (7) vastus medialis (VL) at two-thirds of the distance from the anterior spina iliaca superior to the lateral aspect of the patella; and (8) vastus medialis (VM) at 80% of the distance from the anterior spina iliaca superior to the anterior border of the medial ligament (Figure 1).

Wireless transmitters (ground electrodes) were fixed on the adjacent skin using double-sided tape. The EMG signals were visualized and processed using Noraxon MR3 software (Scottsdale, AZ, USA). Cross-talk and signal-to-noise ratios were visually assessed to ensure signal fidelity before testing. EMG signals were filtered using a band-pass filter of 10–500 Hz and then rectified and smoothed using a 30 ms root mean square (RMS) sliding window. Next, the mean values of the RMS for the pushing and gliding phases for the pushing and standing legs were analyzed in scooter riding. In addition, the area under the RMS curve for each muscle’s EMG for each phase was calculated by multiplying the phase’s mean RMS value by the phase duration. By summing the EMG areas over the pushing and gliding phases for the pushing and standing legs, the area under the RMS curve was calculated over the riding cycle. In running, the mean value of the RMS was calculated for the running cycle.

Wireless IMU sensors (Noraxon) were placed on the pelvis, left and right thighs, shank, and foot in accordance with Noraxon’s guide. The hip angle of the pushing leg was used to identify the scooter riding phases. The pushing phase started at the beginning of hip extension and ended when hip extension reached the maximum angle. The gliding phase started when the hip extension reached the maximum angle and ended at the beginning of hip extension. During the gliding phase, the pushing leg swung forward for preparation for the next push. The video recording was used to clarify the beginning and end of the phases.

The average EMG amplitudes and areas under the RMS curves for the pushing and standing leg were calculated during the pushing and gliding phases for 6–7 cycles starting from the second cycle after crossing the starting line. The symmetry index (*SI*), developed by Robinson et al. [21], was calculated to quantify the EMG symmetry for the pushing and standing legs during scooter riding with the dominant and non-dominant legs as follows.
$$SI=\frac{\left(x_{l}-x_{r}\right)}{0.5\cdot (x_{l}+x_{r})}\cdot 100\%$$
where *SI* is the symmetry index, *x_l_* is the recorded variable for the non-dominant leg, and *x_r_* is the recorded variable for the dominant leg.

### 2.3. Statistics Analysis

Data were tested for normality using the Shapiro–Wilk test. As some data did not meet the criteria for normal distribution, non-parametric statistical methods were selected for further analysis. Mann–Whitney U test was conducted to compare differences in muscle activity (RMS and area under RMS curve) between the dominant and non-dominant legs. Effect size in Mann–Whitney U test was calculated as $r=Z/\sqrt{N}$, where Z is the Z statistics, and N is number of cases. According to Cohen’s [22] guidelines, the r-value was estimated as follows: 0.1 indicated a small effect, 0.3 represented a medium effect, and 0.5 indicated a large effect. Significance was set at *p* < 0.05.

To estimate the sample size, an independent sample means power analysis was conducted. The mean difference and standard deviation for each analyzed muscle were calculated using the RMS values from a pilot study of three children aged 6–8. The analysis indicated that a sample size of 8 would be sufficient for detecting a true effect with 80% power. All data analyses were performed using IBM SPSS Statistics software (v. 22; IBM Corp., Armonk, NY, USA).

## 3. Results

### 3.1. Sprint Running and Riding Duration

The mean sprint running time for 20 m was 4.40 ± 0.25 s. The mean time for 20 m of scooter riding from the convenient side was 6.09 ± 0.76 s (range 4.95–7.64 s), while the mean time for 20 m of scooter riding from the opposite (inconvenient) side was 6.87 ± 1.13 s (range 5.43–8.75). After conducting a Mann–Whitney U test, no statistically significant differences were found between scooter riding from the convenient and inconvenient sides (z = 1.767, *p* = 0.077, r = 0.416).

### 3.2. Muscle Activity during Sprint Running

The individual muscle activity inputs during the 20 m sprint running test are shown in Figure 2. We observed about equal muscle activation between the dominant and non-dominant sides during sprint running for all analyzed muscles (*p* > 0.05) (Figure 2).

### 3.3. EMG during Scooter Riding from the Convenient Side

During scooter riding from the convenient side, a significant difference in muscle activity was observed between the dominant and non-dominant sides during the pushing phase: GL (z = 3.576, *p* < 0.001, r = 0.843), GM (z = 3.576, *p* < 0.001, r = 0.843), BF (z = 2.958, *p* = 0.003, r = 0.697), SM (z = 2.163, *p* = 0.031, r = 0.510), and RF (z = 3.488, *p* < 0.001, r = 0.822). Out of the eight muscles analyzed, five exhibited SI levels that exceeded 50% (Figure 3A). Four of these muscles had a higher RMS in the pushing leg (SI positive), while one muscle had a higher RMS in the standing leg (SI negative). During the gliding phase, four muscles were more active in the standing leg, VL (z = 3.576, *p* < 0.001, r = 0.843), RF (z = 2.782, *p* = 0.005, r = 0.656), VM (z = 3.576, *p* < 0.001, r = 0.843), and TA (z = 2.163, *p* = 0.031, r = 0.510), and one muscle was more active in the pushing leg: GM (z = 2.075, *p* = 0.038, r = 0.489) (Figure 3B).

In all analyzed muscles, there was a significant difference in the area under the RMS curve between the dominant and non-dominant sides throughout the full riding cycle: GL (z = 3.576, *p* < 0.001, r = 0.843), GM (z = 3.576, *p* < 0.001, r = 0.843), BF (z = 2.958, *p* = 0.003, r = 0.697), SM (z = 3.135, *p* = 0.002, r = 0.739), VL (z = 3.135, *p* = 0.002, r = 0.739), RF (z = 3.488, *p* < 0.001, r = 0.822), VM (z = 2.075, *p* = 0.038, r = 0.489), and TA (z = 2.075, *p* = 0.038, r = 0.489). Four of these muscles had a higher RMS in the pushing leg (SI positive), and four had a higher RMS in the standing leg (SI negative) (Figure 3C.)

### 3.4. EMG during Scooter Riding from the Inconvenient Side

During scooter riding from the inconvenient side, a significant difference in muscle activity between the dominant and non-dominant sides during the pushing phase was observed in the three muscles: GL (z = 2.958, *p* = 0.003, r = 0.697), GM (z = 3.488, *p* < 0.001, r = 0.822), and BF (z = 2.782, *p* = 0.005, r = 0.656). The SI of these muscles exceeded 50% (Figure 4A). All three muscles had a higher RMS in the pushing leg (SI negative). In the gliding phase, three muscles were more active in the standing leg (GM (z = 2.163, *p* = 0.031, r = 0.510), BF (z = 2.517, *p* = 0.012, r = 0.593), SM (z = 2.605, *p* = 0.009, r = 0.614)), and three muscles were more active in the pushing leg (VL (z = 3.135, *p* = 0.002, r = 0.739), RF (z = 1.898, *p* = 0.048, r = 0.447) and VM (z = 3.046, *p* = 0.002, r = 0.718) (Figure 4B)).

The area under the RMS curve significantly differed between the dominant and non-dominant sides in five muscles: GL (z = 2.605, *p* = 0.009, r = 0.614), GM (z = 3.576, *p* < 0.001, r = 0.843), BF (z = 3.135, *p* = 0.002, r = 0.739), SM (z = 3.0469, *p* = 0.002, r = 0.719), and RF (z = 2.605, *p* = 0.009, r = 0.614). Four of these muscles had a higher RMS in the pushing leg (SI negative), and one had a higher RMS in the standing leg (SI positive) (Figure 4C).

## 4. Discussion

The purpose of this study was to identify the muscle activation patterns in children’s legs as they rode scooters from the convenient and inconvenient sides. We found up to fivefold differences between the paired muscle activity of the pushing (non-dominant) and standing (dominant) legs, and this difference was highly dependent on the muscle and riding phases. It is possible for an imbalance in the individual muscles to develop. However, the overall asymmetry of the muscle activity between the legs was small and slightly leaned toward the pushing leg side. Comparable patterns, but in the opposite direction, were observed when the children transitioned to the alternate side and performed pushing with the dominant leg, with a tendency for the scooter ride to be completed at a slower pace. These results mainly confirm the expectations about differences in individual muscle loading patterns and suggest that these imbalances are compensated for when driving the scooter from the opposite side.

It is commonly accepted that healthy people symmetrically use their legs during locomotion. In the literature, a difference of 10–15% is used as the threshold to indicate abnormal differences between the limbs [23,24,25]. Such interlimb asymmetries are associated with increased injury risk [26,27,28] and reduced performance [29]. In the present study, we did not find differences in the muscle activation between the legs in the children during their flat sprint running, which indicated balanced contributions by both sides. This is consistent with previous findings of symmetry during walking and jogging [30,31], although some studies reported asymmetry in lower limb muscle activity during walking [13,32]. Daunoraviciene et al. [13] showed that gait in children has no ideal EMG symmetry and that any asymmetry tends more toward the left (non-dominant) side, which may compensate for weakness on that side. These discrepancies between studies may reflect dissimilarities in the participants in the studies. Nonetheless, the overarching basic goal is to achieve low levels of bilateral asymmetry in healthy individuals across both pediatric and adult populations.

Differences between the legs are common during asymmetrical events [33,34,35]. In studies of elite women running the 200 or 400 m sprint in the inner lane of a curved track, significantly higher EMG signals were detected in the left GM than in the right leg muscle, and this pattern persisted throughout the race [34]. Other researchers also found that, during running in football, the maximum EMG amplitude significantly differed between the outer and inner legs, as reflected in the higher EMG activity in the BF and gluteus medius of the inner leg, but there was higher activity in the SM and adductor muscles of the outer leg [33]. When riding a scooter, one foot (usually of the dominant leg) remains positioned on the scooter, and the other foot pushes off. Scooter riding seems to be a one-sided exercise that may be assumed to resemble other sports in which one side of the body predominates, such as golf [36], tennis [37], or fencing [38]. It was a little unexpected that, despite the high muscle activation in the pushing leg compared with the standing leg during the pushing phase, this was mainly reversed during the riding phase, when muscle activation was greater in the standing leg. Hence, in the course of the full exercise (the pushing and riding phases), we observed similar activation in both legs. This finding suggests that scooter riding is a one-sided exercise only for comparisons between individual muscles but not for the comparison between the dominant and non-dominant sides.

Riding a scooter may cause an imbalance between individual muscles, and an imbalance in general may lead to disorders in the musculoskeletal system [9]. However, it remains unclear at what age, for how long, and at what intensity loads must be applied to produce a muscle imbalance. Atkins et al. [39] reported consistent bilateral imbalance in ground reaction forces and that the greatest asymmetries occurred at age 14–16 in football soccer players. There seems to be a “trigger point” during early adolescence when bilateral imbalances may become marked. Tsolakis et al. [38] found asymmetries in the leg muscle morphology in 14–17-year-old adolescents with a 4.4-year fencing training history but not in 10–13-year-old children with a 2.2-year fencing training history. Watanabe et al. [40] concluded that morphological laterality in fencers is elicited by more than 2–3 years of fencing training in juniors. It is possible that a substantial amount of scooter riding is necessary for the development of imbalances, meaning it is unlikely that a limited amount of time spent riding a scooter during leisure time would confer a high risk of bilateral imbalance. Our findings provide some support, in that the 6–8-year-old participants exhibited no bilateral imbalance during the sprint running despite their several years of experience riding scooters. However, this should be considered with caution because the sample size was too small to draw definitive conclusions about long-term adaptations.

The EMG activity, to a great extent, mirrored the opposite side’s finding during driving from the opposite side. It was confirmed that exercising with the contralateral body side induces an antagonistic effect. The observation that the children tended to perform the scooter riding task from the inconvenient side more slowly than from the convenient side is important because it suggests that the children were being more cautious, perhaps because of the novelty of the exercise. Driving from the opposite side seems like a reasonable plan for avoiding asymmetry between the sides of the body, although its safety could be questioned. In this research, we did not intend to investigate the safety of riding from the inconvenient side, but it is likely that a child feels less secure at first. However, all participants in this study were quickly able to use the scooter from the inconvenient side, which suggests that children can easily learn the required skills for safe riding.

### 4.1. Limitations

Despite the statistical confirmation of the sufficient sample size for detecting differences in paired observations, the relatively small overall sample size may reduce the statistical power of the analyses in the present study. Furthermore, various factors including the anthropometry, scooter dimensions, leg position on the scooter, acceleration rate, and riding speed could potentially influence EMG activity, while it is acknowledged that the subject’s experience and skill level can impact bilateral asymmetry as well [41]. Although care was taken in this research to address these issues, it was impossible to control every variable.

### 4.2. Future Directions

This study enhances our understanding of the variations between the two sides of the body in children while engaging in scooter riding and of the potential for this exercise to create muscular imbalances. Subsequent investigations should prioritize examining the impact of scooter riding not only on the lower leg muscles but also on the muscles of the back, which are crucial for maintaining optimal spinal posture. Moreover, it is essential to gain insight into the volume and intensity of scooter riding, which could potentially contribute to the development of muscle imbalances. This knowledge would provide valuable guidance in determining the safe and appropriate dosages of scooter riding exercises, while still allowing children to benefit from this enjoyable and physically engaging activity.

## 5. Conclusions

To summarize, we observed notable differences in the muscle activity patterns of individual muscles between the dominant and non-dominant sides during children’s scooter riding. We also found that these patterns were reversed when the children switched sides on the scooter. These findings suggest that using both legs and switching sides during scooter riding may be a viable strategy for promoting balanced muscular development in children.

## Figures and Tables

**Figure 1 children-10-01064-f001:**
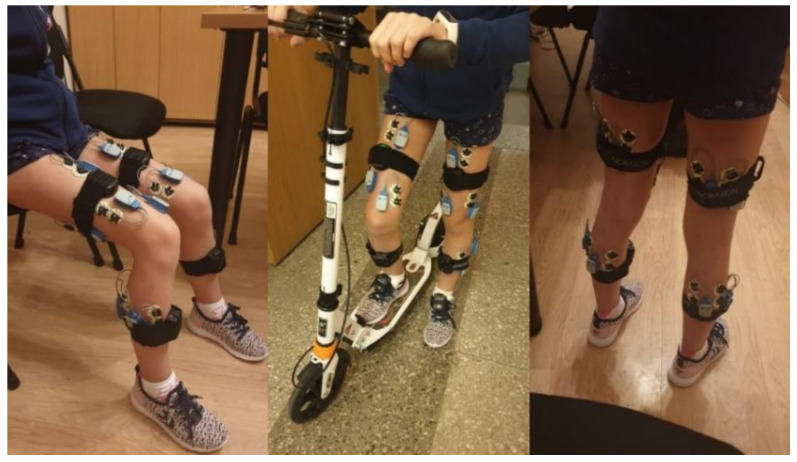
The sensors’ locations.

**Figure 2 children-10-01064-f002:**
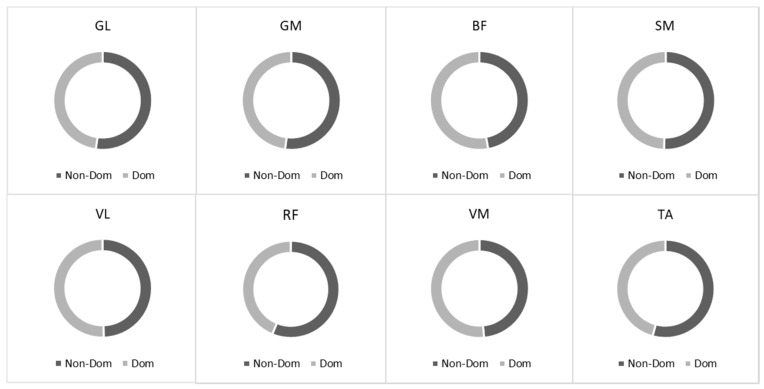
The root mean square (RMS) ratio between dominant and non-dominant sides during running. Notes: GM, gastrocnemius medialis; GL, gastrocnemius lateralis; TA, tibialis anterior; VM, vastus medialis; VL, vastus lateralis; RF, rectus femoris; BF, biceps femoris; SM, semitendinosus.

**Figure 3 children-10-01064-f003:**
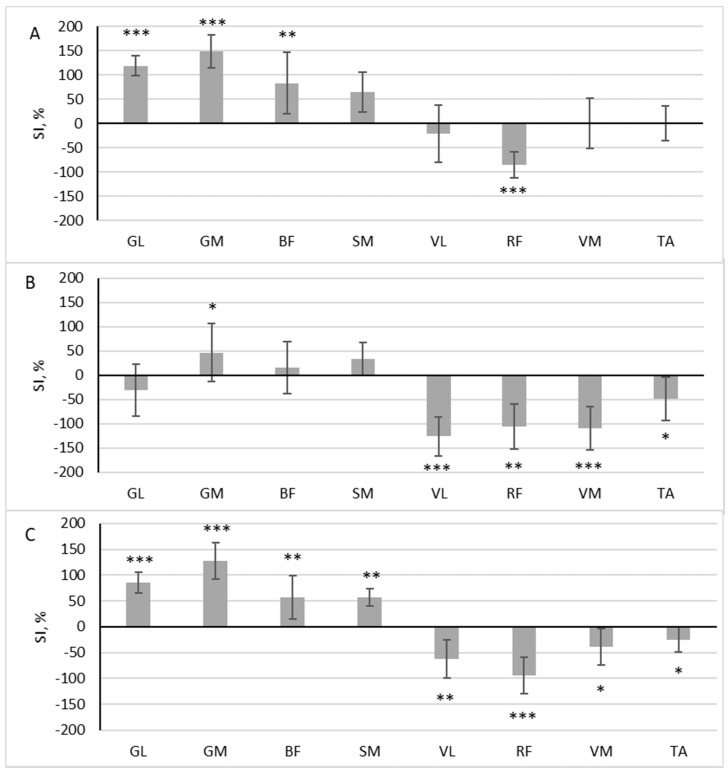
The difference in muscle activity between the dominant and non-dominant sides at pushing phase (**A**), gliding phase (**B**), and full cycle (**C**) during scooter riding from convenient side presented as symmetry index (SI) (mean ± SD). Positive SI indicates higher RMS on non-dominant side, and negative SI indicates higher RMS on dominant side. Notes: GM, gastrocnemius medialis; GL, gastrocnemius lateralis; TA, tibialis anterior; VM, vastus medialis; VL, vastus lateralis; RF, rectus femoris; BF, biceps femoris; SM, semitendinosus. Significant differences in RMS between the dominant and non-dominant sides: * *p* < 0.05, ** *p* < 0.01, and *** *p* < 0.001.

**Figure 4 children-10-01064-f004:**
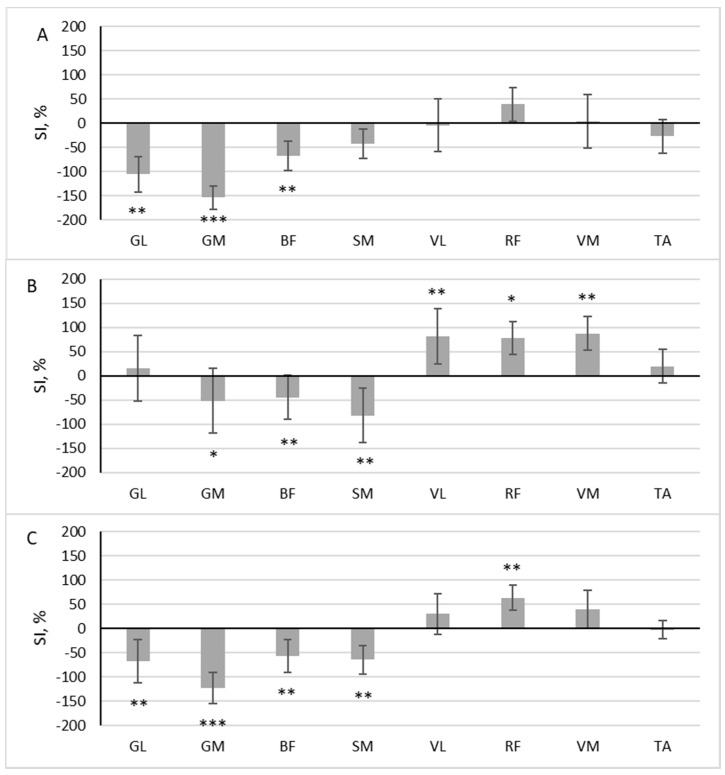
The difference in muscle activity between the dominant and non-dominant sides at pushing phase (**A**), gliding phase (**B**), and full cycle (**C**) during scooter riding from inconvenient side presented as symmetry index (SI) (mean ± SD). Positive SI indicates higher RMS on non-dominant side, and negative SI indicates higher RMS on dominant side. Notes: GM, gastrocnemius medialis; GL, gastrocnemius lateralis; TA, tibialis anterior; VM, vastus medialis; VL, vastus lateralis; RF, rectus femoris; BF, biceps femoris; SM, semitendinosus. Significant differences in RMS between the dominant and non-dominant sides: * *p* < 0.05, ** *p* < 0.01, and *** *p* < 0.001.

**Table 1 children-10-01064-t001:** Age and anthropometric characteristics of the participants.

	Age (Years)	Height (cm)	Weight (kg)	BMI (kg/m^2^)
Boys (*n* = 3)	6.6 (0.6)	128.0 (9.2)	28.5 (6.36)	17.0 (0.5)
Girls (*n* = 6)	7.5 (0.5)	132.0 (5.8)	29.6 (6.8)	17.2 (2.0)
All (*n* = 9)	7.2 (0.7)	130.6 (6.8)	29.3 (6.1)	17.2 (1.7)

Notes: Data are the average (standard deviation).

## Data Availability

Datasets are available on request to the corresponding author.

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
