# Peer review of "Riding a Mechanical Scooter from the Inconvenient Side Promotes Muscular Balance Development in Children"

_children, 2023, doi:10.3390/children10061064_

Round 1
Reviewer 1 Report
A practically relevant study. The authors have collected data and written the article to a good standard and it just needs some fine tuning.
a) The authors should report why there will be a muscular imbalance in children as young as 6, In their sample, they should report on how much experience the children had using scooters as presumably this is a cause. A discussion in the literature on how such imbalances develop and how quickly they develop would be useful.
b) Check if speed was positively skewed with clustering for faster times and a long tail for slower ones. This can be corrected using the inverse transformation (see Box and Cox, 1964).
Box, G. E., & Cox, D. R. (1964). An analysis of transformations. Journal of the Royal Statistical Society: Series B (Methodological), 26(2), 211-243.
c) Make clearer use of effect sizes - the authors use the figures to do this to some extent. Possible report percentage differences in the text. The authors report significance but this is to be expected and something that would be clearly evidenced in larger samples if the study was replicated.
d) This is a pilot study or should be seen as the first study in a series. What is the next study? Before any practical guidelines can be inferred, there needs to be a replication with a large sample size, for example. I am interested in how the authors will take this line of research forwards. The discussion sets out limitations but rather than focus on those, say how the work will be taken forwards and so you accept that there is not a strong focus for application, but will find the evidence in future work.
Written English is fine.
Author Response
Thank you very much for reviewing our manuscript and for the positive comments. The provided feedbacks were valuable and helped us to improve the quality of our manuscript.
The authors should report why there will be a muscular imbalance in children as young as 6, In their sample, they should report on how much experience the children had using scooters as presumably this is a cause.
Reply: The children possessed a range of two to four years' experience in scooter riding. This information was included in revised manuscript. Please see subsection “Participants”.
A discussion in the literature on how such imbalances develop and how quickly they develop would be useful.
Reply: Unfortunately, there is still a lack of clarity regarding the specific age, duration, and intensity of loads necessary to develop a muscle imbalance. This aspect presents an opportunity for further investigation in this research area. Tsolakis et al. (2006) reported that asymmetries in leg muscle morphology in 14- to 17-year-olds with a 4.4-year fencing training history, but not in 10- to 13-year-olds with a 2.2-year history. At the same, Daunoraviciene et al. [11] showed that gait in 8- years old children has no ideal EMG symmetry and that any asymmetry tends more towards the nondominant side possible compensate for weakness on that side. The literature related with these topics are presented in the paragraphs 2-4 of the Discussion.
Check if speed was positively skewed with clustering for faster times and a long tail for slower ones. This can be corrected using the inverse transformation (see Box and Cox, 1964).
Reply: Thank you for suggestion. Indeed, it is important to highlight that we have re-calculated the statistics for all measurements, and the Results section has been partially rewritten. This adjustment was necessary due to the limited sample size, which prompted us to consistently utilize non-parametric statistical methods for data analysis, as certain data did not meet the requirements for normal distribution. It is worth noting that these statistical re-evaluations did not significantly alter the conclusions derived from the previous analyses. However, it is important to mention that no statistically significant differences were observed when comparing the use of the convenient and inconvenient sides for scooter riding (z=1.767, p=0.077, r=0.416), as determined by the Mann-Whitney U test. As a result, the discussion has been accordingly modified to highlight a tendency rather than a significant difference in completing the scooter ride at a slower pace when using the opposite side.
Make clearer use of effect sizes - the authors use the figures to do this to some extent. Possible report percentage differences in the text. The authors report significance but this is to be expected and something that would be clearly evidenced in larger samples if the study was replicated.
Reply: This has been done. As previously mentioned, the Results section has undergone partial revision. Please see pages 5-7.
This is a pilot study or should be seen as the first study in a series. What is the next study? Before any practical guidelines can be inferred, there needs to be a replication with a large sample size, for example. I am interested in how the authors will take this line of research forwards. The discussion sets out limitations but rather than focus on those, say how the work will be taken forwards and so you accept that there is not a strong focus for application, but will find the evidence in future work.
Reply: Thank you for providing your feedback. This study is just the beginning of a series of research regarding the impact of scooter riding on children's muscle development. For our next study, we plan to expand our participant pool and investigate the impact of scooter riding on the back muscles, which are critical in maintaining proper spine posture. No doubts we also would like to bring more clarity regarding the volume and intensity of scooter riding to negatively affect muscle balances.
Reviewer 2 Report
In this study, as a suggestion , "Trainers can identify the weak legs of the athletes and give them scooter riding training with their weak legs" can be added.
This study can be recommended to be conducted among older children (8-13 years old).
In this study, I think that taking the dynamic balance of each leg of the children to the front and back will add more richness. In order to achieve better results, I recommend it to be taken balancing for both leg in a future study like this.

The parentheses around the source numbers on the text are forgotten (15,16), so I wrote it in minor. It is appropriate to reconsider: The 20 m linear sprint test was used as previously described15,16
Author Response
Thank you very much for reviewing our manuscript and for the positive comments.
In this study, as a suggestion, "Trainers can identify the weak legs of the athletes and give them scooter riding training with their weak legs" can be added.
Reply: Thank you very much for suggestion. With the reviewer's consent, we would prefer not to include this particular sentence into the manuscript. The reason being, our study did not investigate the impact of scooter riding on muscle strength, coordination, and posture in athletes. Consequently, this aspect could be reserved for future research.
This study can be recommended to be conducted among older children (8-13 years old).
Reply: We acknowledge the potential relevance of conducting research involving older children. We appreciate your suggestion and will take it into consideration. Thank you.
In this study, I think that taking the dynamic balance of each leg of the children to the front and back will add more richness. In order to achieve better results, I recommend it to be taken balancing for both leg in a future study like this.
Reply: We express our gratitude for the recommendation, and we will make an effort to incorporate it into our subsequent study.
The parentheses around the source numbers on the text are forgotten (15,16), so I wrote it in minor. It is appropriate to reconsider: The 20 m linear sprint test was used as previously described15,16
Reply: Thank you for noticing (corrected in the article).
Reviewer 3 Report
The aim of the study was to identify the muscle activation patterns in children’s dominant and nondominant legs as they rode scooters on the convenient and inconvenient sides. Unofrtunately, I have serious doubts about the reliability of the study. The introduction is very laconic and superficially done, not even exposing the most popular physical activities in children. The number of bibliographic items used in the introduction and discussion shows that the researchers have not fully explored the issue at hand. In addition, what was the basis for the hypothesis at the end of the introduction? This should be completed.
The research sample is far too small. We cannot generalise and draw conclusions on a general or global scale with a study conducted on only 9 children, which is not complicated to implement on a larger sample. From the content of the manuscript, I conclude that the study was conducted in Lithuania, so perhaps it is worth highlighting this fact more, both in the article and in the title of the study. The implementation of the study in a populationally small country somewhat justifies the small research sample. What were the reasons that it was not possible to perform this study on more children? Please add the answer to the manuscript.
The plus point of the article is the methodology, which was chosen correctly. The conclusions support and follow from the research and open up new directions for future research. In addition to the study limitations, please also add the strengths of this study.
I am keeping my fingers crossed for the resolution of all doubts and the eventual publication.
Author Response
We greatly appreciate your valuable feedback and suggestions, which have significantly contributed to the enhancement of our manuscript's quality.
The aim of the study was to identify the muscle activation patterns in children’s dominant and nondominant legs as they rode scooters on the convenient and inconvenient sides. Unofrtunately, I have serious doubts about the reliability of the study. The introduction is very laconic and superficially done, not even exposing the most popular physical activities in children. The number of bibliographic items used in the introduction and discussion shows that the researchers have not fully explored the issue at hand. In addition, what was the basis for the hypothesis at the end of the introduction? This should be completed.
Reply: Several modifications have been made to the Introduction section, specifically addressing the notion that the prevalence of popular physical activities varies across different regions. Notably, children and adolescents worldwide tend to exhibit distinct preferences for team sports and swimming, as highlighted by Hulteen et al. (2017). In the revised Introduction version, we have also incorporated new citations, namely Tannehill et al. (2013), Hulteen et al. (2017), Gonzalo-Skok et al. (2017), and Núñez et al. (2020). Furthermore, we have included additional information emphasizing that our hypothesis is grounded in a longstanding strategy that advocates the implementation of unilateral exercises on the weaker side in order to minimize asymmetry between limbs (Gonzalo-Skok et al. 2017; Núñez et al. 2020). Please see Introduction section.
The research sample is far too small. We cannot generalise and draw conclusions on a general or global scale with a study conducted on only 9 children, which is not complicated to implement on a larger sample. From the content of the manuscript, I conclude that the study was conducted in Lithuania, so perhaps it is worth highlighting this fact more, both in the article and in the title of the study. The implementation of the study in a populationally small country somewhat justifies the small research sample. What were the reasons that it was not possible to perform this study on more children? Please add the answer to the manuscript.
Reply: Before commencing the study, we conducted preliminary measurements on three individuals. These initial measurements revealed substantial differences in the root mean square (RMS) values between observations. In certain cases, these differences reached magnitudes of 5-6 fold, indicating that a sample size of 8 would be adequate to detect differences in paired observations. Subsequently, during the study, numerous statistically significant differences were indeed detected in the comparisons, reaffirming the adequacy of the chosen sample size. To estimate the sample size, an independent-sample means power analysis was conducted. The information regarding sample size counting was slightly rewriten and now is included in statistics analyses paragraph in the revised manuscript version.
However, we acknowledge that the sample size still is relatively small, which in general reduced the statistical power of the analyses. This is acknowledged as a limitation in the revised version of the manuscript.
Furthermore, due to the limited sample size, we consistently employed non-parametric statistical methods for data analysis, as some of the data did not meet the criteria for normal distribution. Consequently, all statistical analyses were conducted accordingly and Results section rewritten considerably. It is noteworthy that these statistical re-evaluations did not alter the conclusions drawn from the previous analyses.
The plus point of the article is the methodology, which was chosen correctly. The conclusions support and follow from the research and open up new directions for future research. In addition to the study limitations, please also add the strengths of this study.
Reply: We appreciate your valuable feedback. This study marks the initial phase of a series of research activities aimed at examining the effects of scooter riding on the muscular development of children. In our forthcoming study, we intend to broaden our participant cohort and explore the influence of scooter riding on the muscles of the back, which play a vital role in maintaining appropriate spinal posture. Additionally, we aim to enhance our understanding of the specific volume and intensity of scooter riding that may contribute to muscle imbalances. This pertinent information has been incorporated into a newly formed paragraph entitled "Future directions" within the manuscript.
Round 2
Reviewer 3 Report
Thank you very much for the additions, which in my opinion have significantly strengthened the manuscript. I keep my fingers crossed for the final success of the publication!